# The Influence of Therapeutic Exercise on Survival and the Quality of Life in Survivorship of Women with Ovarian Cancer 

**DOI:** 10.3390/ijerph192316196

**Published:** 2022-12-03

**Authors:** Sebastián Sicardo Jiménez, Maria Jesus Vinolo-Gil, Ines Carmona-Barrientos, Francisco Javier Martin-Vega, Cristina García-Muñoz, Antonio Rodrigo Guillén Vargas, Gloria Gonzalez-Medina

**Affiliations:** 1Zentrum Medical Clinic, Spain Avenue 112, 41702 Dos Hermanas, Spain; 2Department of Nursing and Physiotherapy, University of Cadiz, 11009 Cadiz, Spain; 3Rehabilitation Clinical Management Unit, Interlevels-Intercenters Hospital Puerta del Mar, Hospital Puerto Real, Cadiz Bay-La Janda Health District, 11006 Cadiz, Spain; 4Research Unit, Department Biomedical Research and Innovation Institute of Cadiz (INiBICA), Puerta del Mar University Hospital, University of Cadiz, 11009 Cadiz, Spain; 5CTS-986 Physical Therapy and Health (FISA), University Institute of Research in Social Sustainable Development (INDESS), 11009 Cadiz, Spain; 6MoveLab Physiotherapy and Training Centre, Condes de Bustillo Street, 46, 41010 Seville, Spain

**Keywords:** neoplasm, cancer, ovarian, therapeutic exercise, survivorship, survival

## Abstract

Background: Ovarian cancer is the most difficult of all gynaecological cancers to treat, mainly due to its late diagnosis. Although exercise interventions have been reported to be safe and beneficial for ovarian cancer, treatment does not include it and is usually a combination of chemotherapy and surgery. It is increasingly common to include exercise as a tool to increase survival in the cancer population. The aim of this study was to determine the influence of therapeutic exercise on survival and the quality of life in survivorship in women with ovarian cancer. Methods: Literature review of clinical trials, reviews and pilot studies compiled in three databases collected in PubMed, PEDro and Scopus. Results: Of the 44 articles found, 10 were selected. Improvements in survival, quality of life and comorbidities associated with ovarian cancer were found with exercise interventions as a treatment tool. Conclusions: Through the application of therapeutic exercise, improvements in survival and quality of life were observed, as well as in the general symptomatology of ovarian cancer. No adverse effects have been found with its use, but future studies with larger sample sizes with more detailed and individualised interventions combined with conventional treatment are needed.

## 1. Introduction

Ovarian cancer, known as the “silent killer” [1], is the most damaging of all gynaecological cancers [2]. It encompasses a group of neoplasms with different clinicopathological and molecular characteristics and prognosis. Although there is a variety of subtypes of ovarian cancer, they are treated as a single disease because when looking at the molecular profile of the cancers, the cells in the fallopian tubes, ovaries and peritoneum all look the same [1,2,3]. In most cases, ovarian cancer is not diagnosed until the disease has entered an advanced stage (more than 70% are not diagnosed until stage III or IV), and this is mainly because its symptoms are usually vague [1]. It is the 13th leading cause of cancer death in the United States. The mortality rate was 6.5 per 100,000 women per year, based on 2015–2019 age-adjusted data [1,4]. In recent years, the trend in ovarian cancer incidence and mortality has decreased due to the increased use of oral contraceptive pills [1].

Ovarian cancer is the second most common malignancy after breast cancer in women over 40 years of age, especially in developed countries, but it is rare in women under the age of 30. The average diagnosis is among women aged 55–64 years [1,4,5], so the risk increases with age and its occurrence rises dramatically after the age of 50.

The prognosis for women who develop ovarian cancer is directly related to the stage of the disease at the time of diagnosis. The stage of cancer at diagnosis determines treatment options and has a major influence on the length of survival [1,4]. The staging of ovarian cancer is similar to the staging of other cancers, which are classified as stages I to IV using the International Federation of Gynaecology and Obstetrics scale as well as the American Joint Committee on Cancer staging system. Generally, the lower the stage number, the less cancer spread there is [2].

The 5-year relative survival rate is only 49%, largely because most populations (58%) are diagnosed with distant stage disease. For women diagnosed with localised disease (Stage I), the 5-year survival rate is 93%, while for those with regional disease (disease has spread to underlying tissues), it is 75%. For women with metastatic disease, the five-year survival rate drops to 30%. Five-year survival is almost twice as high in women younger than 65 years as in women older than 65 years, 61% and 32%, respectively [4,5].

Factors such as the stage of the cancer, the characteristics and subtype of the tumour, and the population’s age, health and preferences must be taken into account when establishing treatment for ovarian cancer [5]. There are multiple treatment options, the most traditional being a combination of chemotherapy and surgery, mainly with the surgical staging of the affected tissue, tumour debulking surgery and subsequent chemotherapy [1].

Including exercise as part of a person’s lifestyle promotes many health benefits. It also applies to people diagnosed with cancer, as well as improving physical well-being and having a positive effect on the different stages of the disease [6]. Exercise interventions are established as safe and beneficial for people undergoing cancer treatment [7] and are positively associated with the reduced severity of the signs and symptoms and adverse side effects of treatment and of developing new cancers [8].

Over the years, different types of exercise interventions have been studied in the cancer population [6], resulting in general recommendations to increase overall physical activity, including specific aerobic or resistance exercises in the cancer care plan. However, it is important to note that exercise prescription requires the consideration of many factors (frequency, duration, intensity, etc.) to positively and safely impact individuals with a cancer diagnosis [7] and should always be supervised by health professionals during its execution, regardless of whether it is performed in clinical settings, at fitness centres or at home [8].

Some of the effects of exercise in cancer populations include: decreasing cancer-related fatigue [7], improving tolerance to treatment, significantly increasing recovery time and sleep, strengthening immune function and cardiovascular system and improving quality of life [8]. In relation to psychological function, reviews show a variable impact of exercise, ranging from positive benefits to inconclusive results with regard to mood, depression and anxiety [7]. However, the reality shows that only 30% of populations include exercise as part of their treatment [9].

The combination of the active form of vitamin D3 with chemotherapy significantly improves the antitumour properties, especially in ovarian omentum cancer cells [1]. Furthermore, epidemiological evidence suggests that adequate vitamin D levels (low vitamin D levels are associated with an increased risk of ovarian cancer) are associated with a lower risk of ovarian cancer and mortality because vitamin D plays an important role in decreasing invasion, abdominal fluid formation and metastasis [10]. Given the relationship between vitamin D and ovarian cancer prevention and, on the other hand, the long-standing evidence that exercise, such as Nordic walking, activates vitamin D [11,12,13], we thought it would be worthwhile to investigate survival data in women with ovarian cancer who exercise.

Examining the literature, we found four previous literature reviews [13,14,15,16], published in 1979, 2015, 2017 and 2020. All conclude that while there is no confirmed evidence regarding survival in women with ovarian cancer and exercise, it should be emphasised that there is a growing body of scientific evidence that has shown that physical activity produces a large number of health benefits that can be achieved in all populations, including those with cancer. It is suggested that physical exercise could improve the comorbidity associated with this pathology (bone density, muscle density, fat and muscle mass index, …) [2]. The interval of years that has elapsed without recent reviews and the somewhat ambiguous results obtained from the existing ones justify updating our knowledge on the subject.

Likewise, no study has been found that demonstrates the influence of therapeutic exercise as a treatment on the survival and quality of life of women with ovarian cancer, without relating them to many other variables associated with the disease [6]. For this reason, the aim of this review is to determine the influence of therapeutic exercise on survival and quality of life in women with ovarian cancer. Our work focuses more on these two variables, although data will be provided on all those that we have found in the study of therapeutic exercise as an active treatment against cancer.

## 2. Material and Methods

### 2.1. Study Design

A literature review was conducted, following the PRISMA 2020 criteria, on clinical trials or reviews no older than 10 years. The search was conducted between July and August 2022. This review is registered in the PROSPERO platform with the code: CRD4202222342007.

The selection of studies was based on PICOS [17]: Participants (P): Studies conducted in humans diagnosed with ovarian cancer. Intervention (I): Studies involving any type of therapeutic exercise as the primary intervention. Comparison (C): With another type of therapeutic intervention or placebo. Outcomes (O): Survival, Quality of life, Cancer-induced fatigue, Sleep quality, Muscle strength, Mental health. Study type (S): clinical trials, systematic reviews or pilot studies. In addition, the Update criteria were taken into account: Studies published within the last 10 years and Language: Studies published in English or Spanish.

Exclusion criteria were: Participants diagnosed with cancers other than ovarian cancer and studies awaiting results and not completed at the date of the literature search.

### 2.2. Search Strategy

The search for scientific papers was carried out in the following databases: Pubmed, Scopus and PEDro. One researcher searched the databases. (S.S.J.).

The keywords were obtained through the Descriptors in Health Sciences (DeCs) [18] and Medical Subject Headings (MeSH) [19]: “exercise therapy”, “ovarian neoplasms”, “cancer”, “ovarian”. The search equations were adapted for use in each database in combination with specific filters, where available (Table 1).

### 2.3. Selection of Studies

After searching the databases, the studies found were screened, following the selection criteria mentioned above. Two researchers performed the screening (S.S.J. and (G.G-M.). In case of doubt, a third investigator (M.J.V.G.) determined whether or not to include the studies.

The selection, with each search equation and database, was as follows:

Pubmed: 1: (exercise therapy [MeSH Terms]) AND (ovarian neoplasms [MeSH Terms]). From this first search, 24 results were obtained, which, after applying the filter of the date of publication of these in the last ten years, left 14 articles.

Of the 14 articles, eight were discarded: three because they were awaiting results, two because they related other types of cancers to ovarian cancer, and three because of the type of study. After discarding the cited studies, six articles remained.

Pubmed: 2: (exercise therapy[MeSH Terms]) AND (cancer, ovarian[MeSH Terms])

From this search we obtained exactly the same results as in search box 1, so no new studies were selected.

PEDro (Simple search): 1: ovarian neoplasms exercise therapy. No results were found.

PEDro (Simple search): 2: ovarian cancer exercise therapy. Four results were found. One of the studies was discarded because it was duplicate, so 3 articles were selected.

SCOPUS: TITLE-ABS-KEY (“exercise therapy” AND (“ovarian cancer” OR “ovarian neoplasms”)) AND PUBYEAR > 2010. After applying the filter of studies carried out in the last ten years, 16 results were obtained, of which only 1 remained. 15 were discarded: two, because they were awaiting results; two, because they related other types of cancers to ovarian cancer; three, because of the type of study; eight, because they were duplicates.

### 2.4. Data Extraction

Once the selection of studies had been completed, two independent reviewers (SSJ and GG-M) extracted the following data from each study: authors, year of publication, study design, sample size and characteristics, main intervention, control intervention, scales or assessment instruments and most relevant conclusions.

### 2.5. Methodological Quality

JBI’s critical appraisal tools assist in assessing the trustworthiness, relevance and results of published papers and were used to measure the methodological quality of the studies.

## 3. Results

### 3.1. Selection of Studies

Out of a total of 44 studies, 10 were selected (Figure 1).

### 3.2. Main Characteristics of the Studies

The selected studies were published between 2011 and 2021. According to the type of study, we can indicate one single-group clinical trial (pre–post treatment), four randomised clinical trials, three reviews (one narrative review and two systematic reviews) and two pilot studies.

The results obtained have been ordered according to the type of study: clinical trial [20,21,22,23,24], review [15,16,25] and pilot studies [26,27]. Subsequently, according to the year of publication, two articles were published in 2011 [20,26], two in 2015 [15,21], three in 2017 [22,25,27], two in 2018 [23,24] and one in 2020 [16].

In terms of sample size, there was a total of 401 women. The total number of articles included in the reviews was 60, being impossible to specify the exact number of studies included in the narrative review by Schofield et al. [25].

The physical exercise interventions carried out in the studies were variable in terms of frequency, intensity, time and type, but most were performed at home and were monitored through telephone calls from professionals.

There is great variability in the scales and instruments used in the studies, the most commonly used being: the SF-36 questionnaire, which is a reliable measure that has been used extensively in cancer populations [22] to assess health-related quality of life (HRQoL); the FACT-O scale, which, after extensive reliability and validity testing, has demonstrated internal consistency and test–retest reliability suitable for measuring quality of life through functional assessment of ovarian cancer treatment [26]; the Pittsburgh Sleep Quality Index (PSQI), which is a self-assessment sleep behaviour questionnaire commonly used in clinical practice to assess sleep rates and sleep duration [23]; Borg Rating of Perceived Exertion (Borg RPE) for perceived exertion during physical exercise, as the heart rate response to exercise may vary among women receiving chemotherapy [27], and body mass index.

All studies conclude that physical exercise appears to be beneficial for women with ovarian cancer, but further research is needed to confirm the extent to which this is true.

A summary of the most relevant results obtained from the studies is shown in Table 2.

### 3.3. Methodological Quality

The methodological quality of the studies included in this review was good or very good. The results can be seen in Table 2.

## 4. Discussion

The present review aims to determine the influence of therapeutic exercise on survival and quality of life in women with ovarian cancer. After analysing the results of the first search of our work and obtaining a total of 44 articles, we consider that, at present, there are few existing studies on ovarian cancer compared with other types of cancer such as breast cancer. Perhaps this is due to the higher incidence of breast cancer in the world’s population, which is more widely studied [4,27].

Not all studies focus solely and exclusively on the survival or quality of life of women with ovarian cancer; some also look at other parameters such as cancer-induced fatigue, sleep quality, mental health and even muscle strength in these women.

There is great variability in the interventions performed, but most are home-based and are supervised by professionals through follow-up telephone calls.

The methodological quality of the studies included in this review was good or very good. This leads us to believe that the studies focused on our objective follow the most relevant guidelines in terms of quality. This will provide more certainty in the discussion of the results. However, the items that were not correctly resolved should not be forgotten, as they are still important.

### 4.1. Synthesis of the Discussion in Terms of Selected Articles

For the review, we will follow a chronological order to see the possible evolution of the findings on therapeutic exercise in this disease over the years, from 2011 to almost the present day.

In 2011, Donnelly et al. [20] conducted a randomised clinical trial that aimed to determine whether a physical activity intervention was feasible and effective for the management of cancer-related fatigue among gynaecological cancer survivors during and after cancer treatments. Secondary outcomes included quality of life, depression and sleep dysfunction, among other variables.

The physical activity performed by the experimental group took place at home for 12 weeks and consisted mainly of walking and strengthening exercises. Similarly, the pilot study by Newton et al. [26] studied the effects of exercising at home, although it consisted only of walking. Both studies [20,26], individualised the frequency of exercise according to the duration of treatment the women under study were undergoing, specifically according to the duration of chemotherapy, and supervision of the subjects was carried out weekly in much the same way in the two studies, through telephone calls or face-to-face consultation with a specialist. This type of exercise has been used by other researchers with the same objective and has produced results along the same lines as ours [28].

Exercise intensity was different in the two studies: While Donnelly et al. [20] opted for moderate intensity as measured by the Borg RPE scale (12–13), Newton et al. [26] individualised the intensity, being low at first and then increasing. The conclusions drawn from these two studies are very similar. Both are in favour of an exercise intervention being beneficial for women with ovarian cancer. Newton et al. [26] conclude that it seems plausible that an individualised walking intervention during chemotherapy can prevent deterioration and reduce the impact of physical symptoms such as lack of appetite, drowsiness, constipation, pain, dry mouth, nausea and weight loss in women with ovarian cancer, improving their physical well-being and quality of life [6].

Donnelly et al. [20] found that a physical activity intervention for gynaecological cancer survivors demonstrates improvements in fatigue, although this improvement may be related to the incorporation of strengthening exercises into the intervention. However, both studies [20,26] justify the need for confirmation of the results in the form of larger and more powerful research.

Retention, adherence and compliance rates in this pilot study were high (over 80%). [26], comparable with feasibility data from previous home-based exercise studies in populations receiving chemotherapy for other cancers [29,30].

Cannioto et al. [15], in 2015, conducted a comprehensive search for epidemiological research focusing on the association between recreational physical activity and risk and survival of epithelial ovarian cancer. The results showed that the protective effect of moderate recreational physical activity on ovarian cancer risk was more consistent among case–control studies than cohort studies. This review included studies that combined moderately active women with completely inactive women in their sample, which makes us doubt whether the results obtained reflect real data to make meaningful comparisons and detect relationships between the two groups. In addition, only three of the studies included in the review [15] associated ovarian cancer survival with exercise, and therefore, no conclusions could be drawn.

In the same year, Mizrahi et al. [21] conducted a 12-week home-based combined physical exercise intervention for women with recurrent ovarian cancer, with low to moderate intensity as measured by the Borg RPE scale (Borg RPE 11–14). Monitoring of the intervention was also conducted [20,26] through weekly phone calls. Two-thirds of women with recurrent ovarian cancer were able to complete a low-to-moderate aerobic exercise programme of 90 min per week without adverse effects, suggesting that physical activity could be used as a treatment option to increase aerobic capacity and strength, accompanied by an improved ability to perform activities of daily living.

In addition, the cardiovascular response helps to decrease fatigue, positively affecting mood, sleep quality and quality of life in women with ovarian cancer. This is comparable with the study by Mock et al. [31], in which a population with advanced breast cancer undergoing chemotherapy treatment who exercised more than 90 min a week improved their fatigue, stress and quality of life, compared with those who were less active [32].

The fact that the design of Mizrahi’s intervention [21] had a duration of 12 weeks and a target of 90 min per week was based on previous research on exercise in oncology during chemotherapy [20,21,32].

Several years later, in 2017, Schofield et al. [25] published a review summarising existing data on the physiological characteristics of ovarian cancer survivors to better understand their level of physical activity during the disease and exercise needs. They concluded that the majority (75%) of ovarian cancer survivors have comorbidities and 15–30% are obese, which may negatively affect the effectiveness and safety of treatment and even the survival of women with ovarian cancer.

On the other hand, during the course of treatment, women suffer a considerable loss of muscle mass, which makes us doubt whether women are still obese or overweight at the end of treatment. One way to help achieve more consistent results in relation to body mass index (BMI) could be to incorporate additional components into the study, such as nutritional counselling, as noted by Donnelly et al. [20] in their study.

Another important finding of the study is that the high levels of interest in physical activity programmes, and some evidence associating physical activity with physical wellbeing, do not correspond with the very high percentages (50–80%) of inactivity or low activity in ovarian cancer survivors. This fact demonstrates the need to adequately and optimally promote the prescription of regular and structured exercise, as well as to correctly inform women affected by this disease about the benefits of exercise, a responsibility that falls to the health care professional.

Similar to the studies [15,20,21,26] already analysed, the review by Schofield et al. [25] challenges the development of future tailored exercise intervention studies and more detailed exercise oncology guidelines.

The randomised controlled trial by Zhou et al. [22] aimed to examine the effect of a six-month aerobic exercise intervention versus a control group on changes in quality of life and cancer-related fatigue in women with ovarian cancer.

Adherence to the intervention was excellent. Some of the features of the intervention resemble a pilot study from the same year by Zhang et al. [27]; both studies [22,27] are 26 weeks long and are monitored through telephone calls, and the physical exercise is performed at home at moderate intensity, although there is a difference in objectifying exercise intensity. While Zhou et al. [22] opt for the Karvonen, Zhang et al. [27] do so with the Borg RPE scale, as we have already seen in other studies [20,21], since this is the scale recommended by the American College of Sports Medicine (ACSM).

Zhou et al. [22] found the six-month home-based, telephone-delivered exercise intervention of 150 min per week of primarily brisk walking to be associated with improved quality of life and a statistically significant improvement on the fatigue scale for the athletes compared with the control group. The women in the experimental group participated in 150 min per week of moderate-intensity aerobic exercise, primarily brisk walking, as recommended by the ACSM and the American Cancer Society. In turn, Zhang et al. and McGrath et al. [6,27] conclude that an exercise programme, in this case, 225 min per week is safe, feasible and acceptable in the advanced ovarian cancer population.

Although previous studies [26] corroborated the feasibility of prescribing exercise to women with ovarian cancer, no previous studies have examined the feasibility of prescribing a higher dose of exercise (more than 150 min per week) in this population, and no studies have examined whether women with advanced ovarian cancer are able to complete a 26-week exercise intervention [27]. Knowing that this study has no control group, and that, of all the studies selected for our review, it is the one with the smallest sample size (n = 10), we have to be cautious with the interpretation of the results.

Subsequently, in 2018, Iyer et al. [24] designed a study on lymphoedema in ovarian cancer that aimed to determine the prevalence of lower extremity lymphoedema in a sample of ovarian cancer survivors, using three different diagnostic methods to assess the effect of a randomised exercise intervention. This study is part of the study by Zhou et al. [22], analysed by us previously.

Despite certain limitations, we include the study [24] in our review because although it focuses on measuring the prevalence of a side effect of ovarian cancer such as lymphoedema, the characteristics of the proposed exercise intervention are similar to those of other studies already reviewed such as the pilot study by Zhang et al. [27] or that of Donnelly et al. [20]; the exercise intervention is performed at home at moderate intensity and consists mainly of brisk walking, but in this case 150 min per week, which is recommended by the ACSM and the American Cancer Society [22]. It is important to emphasise that we must be cautious when interpreting the results of this study because, once again, the results of this study are not always accurate [21,22]: Self-reported questionnaires of a personal nature, such as the SF-36 health questionnaire, are used, so the resulting data can be considered subjective in drawing reliable conclusions.

Nevertheless, as stated in the study itself [24], the self-reported questionnaire had a strong concordance with the baseline assessment by a lymphoedema specialist. In addition, this trial focuses more on the diagnosis, which is necessary to determine the effect of exercise, but less on the intervention itself, which is not very precisely detailed. Again, no adverse effects were found for lower limb lymphoedema.

Returning to the randomised clinical trial by Zhang et al. [27], the aim of the study was to investigate the feasibility of nurse-led home exercise in combination with cognitive behavioural therapy for women with ovarian cancer and cancer-related fatigue in order to test the possible effects on fatigue. This same study also had secondary objectives such as assessing the effects of the intervention on sleep disorders and depression.

For sleep disorders, they used the PSQI, as did Donnelly et al. [20] and Mizrahi et al. [21]. The PSQI is a self-assessed sleep behaviour questionnaire that is commonly used in clinical practice to assess sleep rates and duration. Regarding the results of the study, it was concluded that the intervention model used had benefits in all the objectives pursued, and no adverse effects were found. Contrary to the previously reviewed study by Iyer et al. [24], Zhang and et al. [27] do describe a much more detailed exercise intervention.

We now turn to the 2020 systematic review by Jones et al. [16], who aimed to discover physical activity levels after ovarian cancer diagnosis, determine the relationship between physical activity levels and health outcomes and assess the effect of an exercise intervention on health outcomes for women with ovarian cancer. This review includes five of our selected studies to analyse, namely [21,22,24,26,27], and can therefore be considered the most comprehensive study to date. The relationship between physical activity levels and health outcomes relevant to the populations was evaluated. The most relevant results that we can draw from this review support a positive relationship between higher levels of physical activity and better quality of life, anxiety and depression, although there were only two studies that examined anxiety and depression [31,33]. The number of studies that have studied this relationship is perhaps very low, so we should bear that in mind. Five other studies [34,35,36,37,38] of the 34 analysed by Jones et al. [16] find that higher levels of physical activity after diagnosis are associated with improved quality of life, four studies [39,40,41,42] indicate that they are associated with a reduction in fatigue and two studies [37,41] connect it to improved sleep quality. However, only two studies [43,44] claim that higher levels of physical activity prior to diagnosis are associated with overall survival.

Finally, there is preliminary evidence suggesting a positive relationship between increased physical activity after diagnosis and anxiety, depression, neuropathy, overall survival, side effects, pelvic floor dysfunction and mood disorders [6,16]

We can say that physical activity is relevant to the health outcomes of women with ovarian cancer. Getting women with ovarian cancer to become or remain sufficiently active (including through exercise interventions during or after treatment) has the potential to improve their lives, as physical activity levels compared with prediagnosis decrease and remain low after cancer diagnosis, and most women do not reach WHO-recommended levels of physical activity [16].

Although most of these findings come from cross-sectional studies assessing physical activity more than 12 months after diagnosis, physical activity patterns among women with ovarian cancer represent consistent findings compared with those observed in mixed cancer and breast cancer cohort studies [45,46], as well as in women with gynaecological cancer [47].

### 4.2. Synthesis of the Discussion in Terms of Non-Selected Articles

Finally, we provide data from some studies that were not included in our review because they do not meet all the inclusion and exclusion criteria but which we consider important to point out. El-Sherif et al. [48] concluded in their study that lifestyle, diet and nutrition interventions can prevent and improve survival in ovarian cancer populations. We consider this an important finding, as benefits could possibly be obtained by mixing the study intervention with other physical exercise interventions [22]. It would be interesting to test this in future research to determine whether the scientific evidence is sufficient to support and provide what kind of clinical interventions are best to prevent ovarian cancer and to improve the quality of life of affected women. In fact, there are already two studies that are investigating the effects of a dietary intervention combined with physical activity in women with ovarian cancer. On the one hand, there is a study by Thomson et al. [49] which, although still awaiting results, when completed is likely to be the largest lifestyle intervention trial ever conducted in ovarian cancer survivors. It builds, at present, more widespread clinical recommendations on diet and physical activity after a cancer diagnosis to test the role of lifestyle behaviour change in modifying progression-free survival from a very lethal disease, and so after completion of the two-year protocol intervention, participants are being followed for a further 5 years to collect information on cancer progression and survival, as well as quality of life. Finally, there is a study by Stelten et al. [50] that is a multi-centre, blinded, randomised clinical trial. Data collection began in 2018 and is ongoing. This study will add to the evidence on the potential benefits we have been drawing from our review articles of an exercise and diet intervention in ovarian cancer populations during chemotherapy treatment. If proven effective, a combined exercise and diet intervention for women with ovarian cancer could be used in clinical practice [50]. In summary of the above, the same result was found after analysing exercise interventions for ovarian cancer: no unfavourable effects were reported in any of them.

In only one study of the four that reported on the safety of the interventions in the systematic review by Jones et al. [16], one minor exercise-related adverse event was reported. A fall occurred causing minor cuts and bruises [26].

There is a clear predominance of aerobic exercise with intensities ranging from low-moderate to moderate-vigorous in the interventions performed, concluding that the interventions with the best results are those performed at moderate intensity. Donnelly et al. [9], in addition to aerobic intervention (mainly walking) [28], chose to add strengthening exercises. Others such as Mizrahi et al. [21] or Zhang et al. [23] incorporated resistance exercise, as current guidelines recommend the inclusion of resistance training [50,51]. More recent research also includes other types of interventions such as cognitive behavioural therapy in the study by Zhang et al. [23] or the studies that are still awaiting results [49,52] that mix exercise with diet.

### 4.3. Discussion of the Evolution of the Characteristics of the Studies. Development Standards, Objectivity, Conclusions and Specific Justifications

Over the years, studies have increasingly followed more precise guidelines to establish exercise interventions and to test their influence on survival, quality of life and other factors associated with ovarian cancer.

However, we continue to see little objectivity in areas such as self-reported questionnaire diagnoses. Following Zhang et al. [23], due to the growing number of people living with cancer, there will be an increasing need to find an evidence-based support system to provide interventions that are feasible and justified to help cancer populations alleviate their symptoms [23,53].

The paucity of studies on detailed exercise interventions in women with ovarian cancer may be due to the low 5-year relative survival rate (49.1%) [29] compared with other types of cancer such as breast cancer (90.3%) [27]. This leads to exercise interventions for ovarian cancer that are similar to those for breast cancer, which do have greater support from the scientific literature. Ideally, in the future there should be a physical exercise guide for ovarian cancer populations.

In conclusion, the fact that most exercise interventions are performed at home may probably be due to the fact that the cost is lower than if they are performed in a hospital or a specialised institution. This, in turn, makes it easier for the populations themselves to comply with the intervention, as they would not have to travel, which is a convenience for them. Relating this to the current COVID-19 pandemic situation, according to the study by Chaudhari et al. [54], ovarian cancer and COVID-19 share certain similar molecular and cellular characteristics in their microenvironment, suggesting a possible interaction leading to a poor outcome. In addition, ovarian cancer-associated comorbidities may increase the risk of COVID-19 in women, although COVID-19-related hospitalisations and deaths worldwide are lower in women than in men [54].

Consequently, performing exercise interventions at home today would once again be more than justified, thus preventing women with ovarian cancer from having to travel to hospital; if this were to happen, it would only lead to complications: it would increase the likelihood that women attending hospital for an exercise intervention would end up being admitted to the intensive care unit (ICU), increasing the number of hospitalisations. The COVID-19 pandemic has only added to existing complications, as practical considerations regarding face-to-face visits, ICU bed availability and population concerns also influence on decision-making treatment [55].

### 4.4. Strengths and Limitations of Our Review

The main limitation was the scarcity of existing articles relating physical activity to ovarian cancer, due to the lower incidence and lower survival rate of this disease compared with other types of cancer, such as breast cancer.

No publications have been found of studies aimed solely and exclusively at testing the influence of exercise on the survival and quality of life of women with ovarian cancer without also studying other variables such as cancer-related fatigue, sleep quality and mental health.

The search strategy employed provides much variability in the therapeutic exercise interventions and in the scales and instruments used for the studies, which has made it difficult to interpret the results. In addition, few studies detail in depth the parameters to be followed for exercise interventions. While it is true that clinical trials and reviews that met the established inclusion criteria were included in our review, no meta-analysis was found, and although we are aware that the levels of evidence from pilot studies are lower than those of a randomised clinical trial or systematic review, we have included such studies in the review as we considered them of interest for drawing conclusions.

The total number of women in the study when adding up all the articles in our review can be considered a strength of the study (more than 400 women), as the number of participants in some individual studies was very low.

Although the methodological quality of the included studies is good, limitations are noted in terms of the blinding of both the professionals who prescribe therapeutic exercise and those who analyse the data. This limitation is difficult to overcome, as exercise must be properly and individually prescribed. Still, blinding must be achieved in the analysis of the results.

There is no uniform exercise protocol. This is important as each type of exercise, intensity and duration could affect the results.

## 5. Conclusions

Therapeutic exercise contributes to improved quality of life and survival in women with ovarian cancer. Exercise interventions have been shown to be safe and effective with no adverse events. In addition, exercise contributes to the improvement of cancer symptomatology, with evidence of improvements in cancer-related fatigue, sleep quality and depressive symptoms. Future studies are needed to further investigate more detailed and individualised exercise interventions, specifying what stage of the disease the woman is in; whether the exercise is performed at home or in a specialised centre; and whether it is more beneficial to perform the intervention during or after chemotherapy treatment and specifying the parameters of frequency, intensity, type and time of exercise in order to confirm with certainty the relationship between ovarian cancer and the benefits of a therapeutic exercise intervention.

## Figures and Tables

**Figure 1 ijerph-19-16196-f001:**
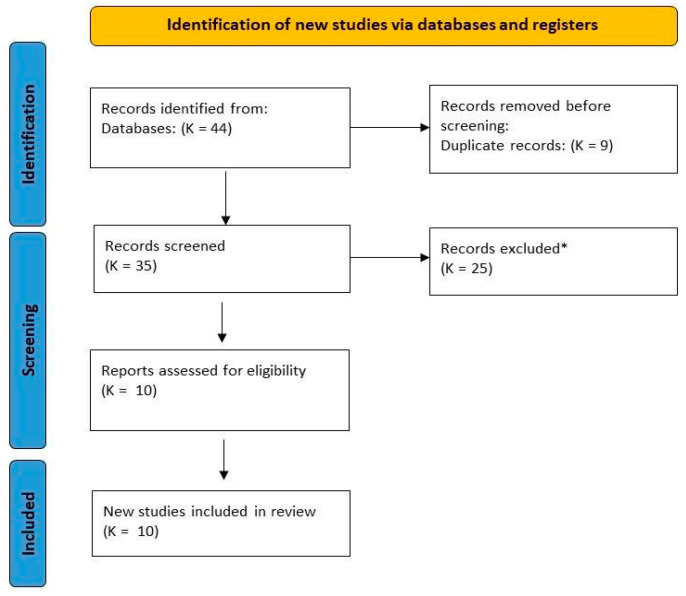
Flow diagram of the studies’ selection process. * excluded for not meeting the inclusion criteria.

**Table 1 ijerph-19-16196-t001:** Search equations in databases.

Search Equations	Database
(exercise therapy[MeSH Terms]) AND (ovarian neoplasms[MeSH Terms]) (exercise therapy[MeSH Terms]) AND (cancer, ovarian[MeSH Terms])	PubMed ^1^
ovarian neoplasms exercise therapy ovarian cancer exercise therapy	PEDro ^2^
“exercise therapy” AND (“ovarian cancer” OR “ovarian neoplasms”)	Scopus ^3^

^1^ PubMed; ^2^ PEDro; ^3^ Scopus.

**Table 2 ijerph-19-16196-t002:** Summary of the most relevant results obtained from the studies.

Study	Year	Type of Study	Sample (n = ) Groups	Interventions	Scales/ Instruments	Conclusions	JBI Evidence Synthesis
Donnelly et al. [19]	2011	Randomised clinical trial	^1^ n = 33 ^2^ EG: n = 16 ^3^ CG: n = 17	^2^ EG: home-based physical activity for 12 weeks. -Frequency: individualised according to the duration of chemotherapy. -Intensity: moderate (Borg RPE 12–13). -Type: walking and strengthening exercises. -Time: individualised (e.g., 10 min at the beginning and then increase). Supervision: 1 initial individual face-to-face session with a physiotherapist + 1 phone call/week for 10 weeks + final face-to-face consultation at week 12 and 2 months of follow-up phone calls. ^3^ CG: standard care. Same phone calls as ^2^ EG. No advice on how to change their physical activity levels during the study period.	^5^ MFSI-SF ^6^ FACIT-F ^7^ FACT-G ^8^ BDI-II ^9^ PANAS ^10^ BMI ^11^ 12MWT ^12^ Borg RPE ^13^ PSQI ^14^ 7 Day PAR	A physical activity intervention for gynaecological cancer survivors demonstrates improvements in fatigue, but confirmation in the form of a larger, more powered randomised clinical trial is warranted.	11/13
Mizrahi et al. [20]	2015	Single group clinical trial (pre/post)	^4^ GP/P: n = 30	^4^ GP/P: physical exercise carried out at home. -Frequency: 3–4 times/week for 12 weeks. -Intensity: Low-Moderate (Borg RPE 11–14). -Resistance: 3 sets per exercise of 10 repetitions. -Type: Combination of aerobic (walking, cycling, swimming), resistance (resistance band, body weight exercises), core stability (floor, stability ball exercises) and balance exercises. -Time: 10–40 min/session, 90 min/wk. - Supervision: 1 face-to-face session + 1 telephone session/week.	^15^ IPAQ ^13^ PSQI ^16^ SPHERE ^17^ SF-36 ^18^ FACT-O ^19^ FACT-Ntx ^12^ Borg RPE	The results indicate that two-thirds of women with recurrent ovarian cancer were able to complete a low-to-moderate aerobic exercise programme of 90 min per week, with no adverse events reported. Randomised control studies are required to confirm the benefits of exercise reported in this study.	8/9
Zhou et al. [21]	2017	Randomised clinical trial	^1^ n = 144 ^2^ GE: n = 74 ^3^ GC: n = 70	^2^ EG: physical exercise performed at home. -Frequency: 26 weeks. -Intensity: Moderate (target heart rate range based on the Karvonen method for moderate to vigorous intensity. The study provided heart rate monitors). -Type: Aerobic (mainly brisk walking) -Time: 150 min/week. -Supervision: 1 phone session/week with a ^20^ACSM-certified cancer exercise coach. ^3^ CG: 26 weeks, information related to ovarian cancer survivorship. -Supervision: 1 phone session/week with a 21WALC staff member. ^21^ WALC staff member.	^22^ HRQOL ^23^ FACT-F ^17^ SF-36 ^24^ MCS ^25^ PCS	A six-month, home-based, telephone-delivered exercise intervention, consisting primarily of brisk walking, was shown to be associated with improved physical health-related quality of life in women with ovarian cancer. For this reason, health professionals should recommend and refer women diagnosed with ovarian cancer for physical exercise programmes in the clinic or in the community.	9/13
Zhang et al. [26]	2018	Randomised clinical trial	^1^ n = 72 ^2^ EG: n = 36 ^3^ CG: n = 36	^2^ GE: physical exercise performed at home + ^26^ TCC. -Frequency: 3–5 times/week. -Intensity: 40–75% of ^27^ FCM. Type: warm-up, aerobic, muscle strength, resistance training and stretching, deep relaxation and cool down. -Time: 25–60 min/session. -Supervision: 1 telephone session/week conducted by experienced nurses. ^26^ CBT: via internet. 1x/week for 12 consecutive weeks. Duration: 1 h/session. ^3^ CG: no special care. Only medication education, balanced diet recommendations and health education on ovarian cancer chemotherapy.	^28^ PFS ^29^ SDS ^13^ PSQI ^30^ ANOVA	The results indicate that exercise plus nurse-delivered home-based ^26^ CBT has measurable benefits in helping women with ovarian cancer decrease cancer-related fatigue and depressive symptoms and improve their sleep quality.	11/13
Iyer et al. [23]	2018	Part of a randomised controlled trial.	^1^ n = 95 ^2^ EG: n = 50 ^3^ CG: n = 45	^2^ EG: physical exercise performed at home. -Frequency: 26 weeks. -Intensity: moderate (target heart rate range based on the Karvonen method for moderate to vigorous intensity. The study provided heart rate monitors). -Type: Aerobic (mainly brisk walking). Time: 150 min/week. -Supervision: 1 telephone session/week with a ^20^ ACSM certified cancer exercise trainer. ^3^ CG: 26 weeks, information related to ovarian cancer survivorship. -Supervision: 1 phone session/week with a ^21^ WALC staff member.	Self-Report questionnaire. Electronic periometer. Assessment by a physiotherapist specialising in lymphoedema.	With a potential prevalence of lower limb lymphoedema (LLL) of up to 40%, further evaluation of diagnostic methods is required to better characterise this side effect of ovarian cancer treatment. No adverse effects of exercise on LLL were found. Further research is needed to evaluate predictors of LLL and the effects of exercise on LLL to develop effective physical activity recommendations for women with ovarian cancer.	10/13
Cannioto et al. [14]	2015	Systematic Review	Number of studys: 26	A comprehensive literature search was conducted through PubMed for epidemiological investigations focusing on the association between ^31^ RPA and epithelial ovarian cancer risk and survival.	Total number of hours or ^32^ MET hours of activity performed/week. ^10^ BMI	Due to the limitations found in the research, emphasis should be placed on the larger body of scientific literature, which has shown that leading a physically active lifestyle results in a wide range of benefits.	6/11
Schofield et al. [24]	2017	Narrative Review	Number of studys: not specified	A review was conducted of the current literature in PubMed, MEDLINE, CINAHL and SPORTDiscus databases of English-language articles published between January 1970 and December 2016 on the physiological status of ovarian cancer survivors.	Physiological status such as treatment-related adverse effects, comorbidities, body weight and composition, physical fitness and function, and participation in physical activity were defined. ^10^ BMI	Ovarian cancer survivors may benefit from physical activity and exercise interventions aimed at addressing the harms and changes in patients’ physiological status due to disease and treatment. However, gaps in knowledge regarding physiological characteristics across the survivorship spectrum currently remain and challenge the development of tailored exercise intervention studies and exercise oncology guidelines.	6/11
Jones et al. [15]	2020	Systematic Review	Number of studys: 34	A systematic literature search was conducted using different databases such as PubMed, EMBASE, Scopus and CINAHL until 31 December 2019, to find out the possible benefits of physical activity and exercise in women with ovarian cancer.	^6^ FACIT-F ^7^ FACT-G ^13^ PSQI ^17^ SF-36 ^18^ FACT-O ^33^ EORTC QLQ-30 ^30^ s sit-to-stand ^34^ 6 MWT ^35^ MSAS-PHYS	The results obtained suggest that physical activity is relevant to the health of women with ovarian cancer, providing benefits and improvements in their lives. Furthermore, future work is needed to evaluate specific exercise interventions to ensure that the results obtained in this review can be translated into improved ovarian cancer care.	10/11
Newton et al. [25]	2011	Pilot study	^4^ GP/P: ^1^ n = 17	^4^ GP/P: exercise performed at home. -Frequency: individualised according to the length of chemotherapy. Mostly 4 days/week. -Intensity: individualised (e.g., low at first, then increasing). -Type: Aerobic (walking). -Time: individualised. Mostly 30 min/session. - Supervision: 1 session/week with a specialist (face-to-face or by phone). (in person or by telephone).	^18^ FACT-O ^34^ 6 MWT ^35^ MSAS-PHYS ^36^ HADS Likert Scale	Significant improvements in physical functioning, symptoms, physical well-being and quality of life were found, suggesting that a walking intervention for women receiving chemotherapy for ovarian cancer is safe, feasible and acceptable and could be used to develop future studies.	8/9
Zhang et al. [22]	2017	Pilot study	^4^ GP/P: ^1^ n = 10	4GP/P: home exercise. -Frequency: 26 weeks. -Intensity: moderate (Borg RPE 12–15). -Type: Aerobic (mainly walking, although cycling and cardiovascular training equipment were allowed). -Time: 225 min/week. -Supervision: - Week 1–6: face-to-face session/week with a certified trainer. - Week 7–26: face-to-face session/month with a certified trainer.	^10^ BMI ^12^ Borg RPE Triaxial accelerometer ActiGraph GT3X Fitbit Zip	The results suggest that prescribing an exercise programme of 225 min per week to women with advanced ovarian cancer is feasible, safe and acceptable, although the need for a definitive clinical trial to evaluate the potential therapeutic effects of exercise on disease-related symptoms and endpoints among women with ovarian cancer is highlighted.	9/9

^1^ n: Number of participants; ^2^ EG: Experimental Group. ^3^ CG: Control Group; ^4^ GP/P: pre/post Group; ^5^ MFSI-SF: Fatigue Symptom Inventory-Short Form; ^6^ FACIT-F: Functional Assessment in Chronic Illness Therapy-Fatigue subscale; ^7^ FACT-G: Functional Assessment of Cancer Therapy-General Scale; ^8^ BDI-II: depression; ^9^ PANAS: Positive and Negative Affect Schedule; ^10^ BMI: Body Mass Index; ^11^12MWT: 12 min Walking Test; ^12^ Borg RPE: Borg Rating of Perceived Exertion; ^13^ PSQI: Pittsburgh Sleep Quality Index; ^14^ 7 Day-PAR: 7 Day Physical Activity Recall; ^15^ IPAQ: International Physical Activity Questionnaire; ^16^ SPHERE: Somatic Psychological Health Report; ^17^ SF-36: Short Form-36. ^18^ FACT-O: Functional Assessment Cancer Therapy-Ovarian; ^19^ FACT-Ntx: Neurotoxicity; ^20^ ACSM: American College of Sports Medicine; ^21^ WALC: Women’s Activity and Lifestyle Study in Connecticut; ^22^ HRQOL: Health-Related Quality of Life; ^23^ FACT-F: Functional Assessment of Cancer Therapy-Fatigue; ^24^ MCS: mental component summary; ^25^ PCS: physical component summary; ^26^ TCC: Terapia Cognitivo Conductual; ^27^ FCM: Frecuencia Cardíaca Máxima; ^28^ PFS: Piper Fatigue Scale; ^29^ SDS: Self-Rating Depression Scale; ^30^ ANOVA: analysis of variance; ^31^ RPA: Recreational Physical Activity; ^32^ MET: Metabolic Equivalents; ^33^ EORTC QLQ-30: European Organisation for the Research and Treatment of Cancer Quality of Life Questionnaire Version 3.0; ^34^ 6 MWT: 6 min Walk Test; ^35^ MSAS-PHYS: Memorial Symptom Assessment Scale—Physical Symptoms Subscale; ^36^ HADS: Hospital Anxiety and Depression Scale.

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
