# Peer review of "The Influence of Therapeutic Exercise on Survival and the Quality of Life in Survivorship of Women with Ovarian Cancer"

_ijerph, 2022, doi:10.3390/ijerph192316196_

Round 1
Reviewer 1 Report
- The paper title should reflect that this is a systmetic review. On first glance, it reflects a research article.
- Proper paragraphs need to be included in the introudction. Simply writing on short sentence per paragraph is not recommended.
- Theoretical background, basics, and research steps are well-followed. Howeover, more recent references are available in the literature (e.g., McGrath D, O'Halloran P, Prue G, Brown M, Millar J, O'Donnell A, McWilliams L, Murphy C, Hinds G, Reid J. Exercise Interventions for Women with Ovarian Cancer: A Realist Review. Healthcare (Basel). 2022 Apr 13;10(4):720. doi: 10.3390/healthcare10040720. PMID: 35455897; PMCID: PMC9024745.) This reference also presents a proper paper organization in this domain.
- Generally, references are up to date and properly cited as well.
- The novelty of the paper need to be highlighted with good number of references.
- Paper focus is on an interesting topic and the use of PRISMA criteria is sound.
Author Response
Dear reviewer,
Thank you very much for your comments and the review of the manuscript. They will undoubtedly be very helpful to improve it. Below, you will find the answers to your queries and comment that needed to be revised.
Response reviewer 1:
- The paper title should reflect that this is a systmetic review. On first glance, it reflects a research article.
Response: Following your indications the information has been included in the title. Thank you
- Proper paragraphs need to be included in the introduction. Simply writing on short sentence per paragraph is not recommended.
Response: We have modified the introduction, now we hope it is correct. Thank you
- Theoretical background, basics, and research steps are well-followed. Howeover, more recent references are available in the literature (e.g., McGrath D, O'Halloran P, Prue G, Brown M, Millar J, O'Donnell A, McWilliams L, Murphy C, Hinds G, Reid J. Exercise Interventions for Women with Ovarian Cancer: A Realist Review. Healthcare (Basel). 2022 Apr 13;10(4):720. doi: 10.3390/healthcare10040720. PMID: 35455897; PMCID: PMC9024745.) This reference also presents a proper paper organization in this domain.
Response: Thank you very much for your recommendation; it was really helpful and the proposed literature has been included.
- Generally, references are up to date and properly cited as well.
Response: Thank you very much for your comment. In addition to the previously mentioned other references proposed by reviewers have been added.
- The novelty of the paper need to be highlighted with good number of references.
Response: Proper changes were performed and to justify the novelty, your proposed reference of McGrath et al. was used. Thank you once again, you suggestion have been really useful.
- Paper focus is on an interesting topic and the use of PRISMA criteria is sound.
Response: We really appreciate your comments and also your previous suggestions, that improve our manuscript. Thanks
Reviewer 2 Report
1. Study rationale has been written properly. Introduction contain a lot of paragraphs with single sentences.
2. Exclusion criteria – what about grey literature?
3. I have serious concerns regarding the search term used. It looks like the authors didn’t search for manuscripts properly. Why are the synonyms not used in the search? Why is proper keyword search not done in PEDro . The keywords have to be selected according to PICO
4. Was the references in the selected studies manually searched for finding out more studies
5. The PRISMA flow diagram is incomplete
6. Why is risk bias assessment not done?
7. The results are not in line with the study's objective.
8. The discussion has to change according to the results after making corrections to the results
Author Response
Dear reviewer,
Thank you very much for your comments and the review of the manuscript. They will undoubtedly be very helpful to improve it. Below, you will find the answers to your queries and comment that needed to be revised.
- Study rationale has been written properly. Introduction contain a lot of paragraphs with single sentences.
Response: To avoid this problem part of the introduction have been modified. We hope now it is correct. Thanks for your comment.
- Exclusion criteria – what about grey literature?
Response: Grey literature was not considered in our search due to next reasons: i) Grey literature could present significant risk of bias as indicated the study of Adams et al. (1); ii) We built a wide search to avoid losing potential records, but if we conducted this search within grey literature the amount of literature would be unaffordable, so we decided to limit our search to conventional databases.
- I have serious concerns regarding the search term used. It looks like the authors didn’t search for manuscripts properly. Why are the synonyms not used in the search? Why is proper keyword search not done in PEDro. The keywords have to be selected according to PICO
Response: Sorry about the inconvenience. After carrying out small pilot searches, we observed that the results could be scarce within databases. For this reason, we decided to conduct a wider search without limiting by specific terms, avoiding the loss of potential results, despite the fact that this would entail a broader screening and selection processes.
- Was the references in the selected studies manually searched for finding out more studies.
Response: In order to conduct a reproducible search, we only performed the selection process within databases. A manual search or gray literature search was not carried out
- The PRISMA flow diagram is incomplete
Response: The boxes in the flow chart, which are not applicable, have been excluded. This is based on: From: Page MJ, McKenzie JE, Bossuyt PM, Boutron I, Hoffmann TC, Mulrow CD, et al. The PRISMA 2020 statement: an updated guideline for reporting systematic reviews. BMJ 2021;372:n71. doi: 10.1136/bmj.n71. please let us know if you think any further information should be added and we will include it. Thank you.
- Why is risk bias assessment not done?
Response: JBI’s critical appraisal tools assist in assessing the trustworthiness, relevance and results of published papers has been added to the current manuscript to measure the methodological quality of the studies. Thank you for this suggestion, we think this really helped to improve our manuscript.
- The results are not in line with the study's objective.
Response: New findings have been added to the review and we think they are in line with the objective. Thank you for your comment.
- The discussion has to change according to the results after making corrections to the results
Response: Thank you for your comment. The discussion has been reviewed, adding new information and references. However, no major changes have been added to the discussion because other reviewers reported that it was sound, so no major changes were made. Sorry about the inconvenience.
Reviewer 3 Report
Dear Authors,
The manuscript presented for review is a very interesting review on the effect of exercise on survival and quality of life in subjects with ovarian cancer. The article is generally well written, the methodology does not raise any major concerns, the results presentation needs improvement, the discussion is well conducted.
Please find detailed comments below.
Abstract:
Lines 20-24: Please remove Simple Summary section
Line 25: Please replace “deadly cancer” with another term
Line 26-27: Please rewrite this sentence to make it more logic
Line 28: Should be: “The aim of this study was”
Line 38: Please replace “neoplasia” with “neoplasm”
Introduction:
Line 41: Please replace “deadliest” with another term
Lines 45-53: It all should be in one paragraph
Lines 76-80: This paragraph is interesting but doesn’t fit here – please remove
Line 82-84: Please rewrite this sentence
Line 89: Change “population” to “patients” or “patients population”
Line 104: please explain how “exercise activates vitamin D”. You also could add the deleted paragraph from lines 76-80 here. There’s also one newly published study concerning the effect of exercise on vitamin D metabolite level in cancer remission stage – it could be added here:
CzerwiÅ„ska-Ledwig, O.; Vesole, D.H.; Piotrowska, A.; Gradek, J.; Pilch, W.; Jurczyszyn, A. Effect of a 6-Week Cycle of Nordic Walking Training on Vitamin 25(OH)D3, Calcium-Phosphate Metabolism and Muscle Damage in Multiple Myeloma Patients–Randomized Controlled Trial. J. Clin. Med. 2022, 11, 6534. https://doi.org/10.3390/jcm11216534
Line 117: word “found” is repeated in one sentence
Line 117-120: this sentence should be rewritten
Material and methods:
Please specify if search was performed by one researcher, more than one researcher or the same search was made by more than one researcher and the results were compared?
Results:
Line 211: Table 1 is the table with search equations – please change current table number to Table 2. Also, table with the results is missing – please add it!
Discussion:
This part is generally well written but it’s difficult to evaluate this section without the table containing the results.
Study limitations:
I would also add a comment that there is no uniform exercise protocol – it’s worth emphasizing as each type of exercise, intensity, duration could affect the results.
Author Response
Dear reviewer,
Thank you very much for your comments and the review of the manuscript. They will undoubtedly be very helpful to improve it. Below, you will find the answers to your queries and comment that needed to be revised.
Response reviewer 3:
Dear Authors,
The manuscript presented for review is a very interesting review on the effect of exercise on survival and quality of life in subjects with ovarian cancer. The article is generally well written, the methodology does not raise any major concerns, the results presentation needs improvement, the discussion is well conducted.
Please find detailed comments below.
Response: Thank you for your time in reviewing our manuscript.
Abstract:
Lines 20-24: Please remove Simple Summary section
Line 25: Please replace “deadly cancer” with another term
Line 26-27: Please rewrite this sentence to make it more logic
Line 28: Should be: “The aim of this study was”
Line 38: Please replace “neoplasia” with “neoplasm”
Response: The proposed changes for the abstract have been implemented. Thank you.
Introduction:
Line 41: Please replace “deadliest” with another term
Lines 45-53: It all should be in one paragraph
Lines 76-80: This paragraph is interesting but doesn’t fit here – please remove
Line 82-84: Please rewrite this sentence
Line 89: Change “population” to “patients” or “patients population”
Line 104: please explain how “exercise activates vitamin D”. You also could add the deleted paragraph from lines 76-80 here. There’s also one newly published study concerning the effect of exercise on vitamin D metabolite level in cancer remission stage – it could be added here:
CzerwiÅ„ska-Ledwig, O.; Vesole, D.H.; Piotrowska, A.; Gradek, J.; Pilch, W.; Jurczyszyn, A. Effect of a 6-Week Cycle of Nordic Walking Training on Vitamin 25(OH)D3, Calcium-Phosphate Metabolism and Muscle Damage in Multiple Myeloma Patients–Randomized Controlled Trial. J. Clin. Med. 2022, 11, 6534. https://doi.org/10.3390/jcm11216534
Line 117: word “found” is repeated in one sentence
Line 117-120: this sentence should be rewritten
Response: The proposed changes for the introduction have been implemented. Thank you.
Material and methods:
Please specify if search was performed by one researcher, more than one researcher or the same search was made by more than one researcher and the results were compared?
Response: The proposed changes in the material and method section have been implemented. Thank you.
Results:
Line 211: Table 1 is the table with search equations – please change current table number to Table 2. Also, table with the results is missing – please add it!
Response: Thank you very much for your comment. It has helped us to improve our manuscript considerably.
Discussion:
This part is generally well written but it’s difficult to evaluate this section without the table containing the results.
Response: The use of the table in the results section has reinforced the discussion. Thank you for your useful suggestion.
Study limitations:
I would also add a comment that there is no uniform exercise protocol – it’s worth emphasizing as each type of exercise, intensity, duration could affect the results.
Response: The indicated limitation has been added. Thank you
Reference:
- Adams, J., Hillier-Brown, F.C., Moore, H.J. et al. Searching and synthesising ‘grey literature’ and ‘grey information’ in public health: critical reflections on three case studies. Syst Rev. 2016; 5:164. https://doi.org/10.1186/s13643-016-0337-y
Round 2
Reviewer 3 Report
Table 2 has been added as a supplementary material - I suggested to add it as a part of the manuscript. Aditionally, this table is still listed in the text. Please either remove the reference to the Table 2 in the text or add the table to the manuscript.
Author Response
Response to Reviewer 3 Comments
Point 1: Table 2 has been added as a supplementary material - I suggested to add it as a part of the manuscript. Aditionally, this table is still listed in the text. Please either remove the reference to the Table 2 in the text or add the table to the manuscript.
Response 1: We have included the table within the manuscript. We are very grateful for your comments.